# The Use of Principal Component Analysis for Reduction in Sleep Quality and Quantity Data in Female Professional Soccer

**DOI:** 10.3390/s25010148

**Published:** 2024-12-30

**Authors:** Eider Barba, David Casamichana, Pedro Figueiredo, Fábio Yuzo Nakamura, Julen Castellano

**Affiliations:** 1Real Sociedad Institute, Real Sociedad de Fútbol S.A.D., 20170 Donostia-San Sebastian, Spain; david.casamichana@realsociedad.eus; 2GIKAFIT Research Group, University of the Basque Country (UPV/EHU), 01007 Vitoria-Gasteiz, Spain; julen.castellano@ehu.eus; 3Physical Education Department, College of Education, United Arab Emirates University, Al Ain P.O. Box 15551, United Arab Emirates; pfigueiredo@uaeu.ac.ae; 4Research Center in Sports Sciences, Health Sciences and Human Development (CIDESD), University of Maia, 4475-690 Maia, Portugal

**Keywords:** multivariate analysis, sleep, smart ring, soccer

## Abstract

The main aim of the present study was to uncover multivariate relationships between sleep quantity and quality using principal component analysis (PCA) in professional female soccer players. A second aim was to examine the extent to which objective sleep quantity and quality variables can discriminate between perceived sleep. Ten objective sleep variables from the multisensory sleep-tracker were analyzed. PCA was conducted on the sleep variables, and meaningful principal components (PCs) were identified (eigenvalue > 2). Two sleep PCs were identified, representing the ‘quantity of sleep’ (quantity PC: eigenvalue = 4.1 and variance explained = 45.1%) and the ‘quality of sleep’ (quality PC: eigenvalue = 2.4 and variance explained = 24.1%). Cluster analysis grouped the players’ sleep into three types: long and efficient, short and efficient, and long and inefficient; however, no association was found between the perceived sleep and the sleep clusters. In conclusion, a combination of both quantity and quality sleep metrics is recommended for sleep monitoring of professional female soccer players. Players should undergo a training process to improve self-assessment of sleep quality recorded from a subjective questionnaire, contrasting the perceived information with the sleep quality recorded objectively during a defined period in order to optimize the validity of their perceptions. The aim is to optimize the validity of their perceptions of sleep quality.

## 1. Introduction

Sleep is a crucial biological necessity for human well-being and health [1]. In the context of sport, sleep provides several important psychological and physiological functions that are fundamental to the recovery process [2,3]. For this reason, the need for adequate sleep has become one of the fundamental pillars of recovery from the fatigue caused by training and competition [3,4,5]. Given the growing concern about athletes’ and soccer players’ sleep, studies examining sleep interventions have increased exponentially in recent years [6].

Different techniques and technologies have emerged in the sports setting to measure sleep, such as questionnaires or scales, polysomnography (PSG), and wearable technology [7]. Questionnaires and scales offer a simple and low-cost method for obtaining subjective information. The gold standard technology for measuring sleep is the PSG; however, this technology is expensive and less convenient [8], which often limits its use to a restricted number of sessions or weeks of monitored use [9]. Good alternative methods to PSG include wearable technological devices such as actigraphy, smartwatches, smart rings, wristbands, and fitness trackers [10,11]. So far, the most widely used technology has been actigraphy, as it is a validated and reliable technology that is simple and easy to use daily [8]. However, other technologies have been gaining momentum as they enable devices to automatically detect sleep in real time, providing sleep metrics the following morning through a digital platform (e.g., a smartphone app), which makes them easy to use in everyday life [8,10]. It is important to note that these commercial devices show significant variations between them [11], and their accuracy can vary depending on the device and the specific sleep metric being measured. Despite these limitations, the convenience, cost-effectiveness, and ability to monitor sleep in natural settings make wearable sensors a valuable tool for sleep analysis in sports, especially when compared to more invasive and resource-intensive methods like PSG or more subjective methods such as questionnaires.

In the specific case of sleep, numerous sleep variables are generated by wearable devices. To facilitate efficient and objective decision-making regarding fatigue-fitness management and to optimize the athlete’s performance and health, it is essential to reduce the number of variables to manage [12,13]. Moreover, many of these variables are often correlated, leading to multicollinearity and redundancy in the dataset [13]. PCA serves as a powerful reductionist method that simplifies the complexity inherent in the vast array of metrics collected by sensors [14]. By transforming the original correlated variables into a smaller set of uncorrelated components, PCA not only preserves the essential information but also enhances interpretability [14]. This reduction allows practitioners to focus on the most significant factors influencing quality and performance, thereby streamlining the analysis process and improving the overall effectiveness of fatigue management strategies [14]. Principal component analysis (PCA) has been used in previous studies to identify the relationships between multiple training load variables, which can facilitate feature selection to inform decision-making [15,16].

Another way to measure sleep is by using questionnaires or self-reported scales. These subjective measures, being inexpensive and simple to implement compared to objective measures, may offer an alternative for monitoring sleep in athletes [17]. However, it should be noted that several studies have shown that athletes tend to overestimate the quality and quantity of their actual sleep [18,19,20]. Nevertheless, determining the extent to which the objective sleep variables relate to the subjective sleep is important for the holistic management of the sleep process.

Therefore, the main objective of the present study was to use principal component analysis for the reduction in sleep quality and quantity data in female professional soccer. A second objective was to examine the extent to which objective variables of sleep quantity and quality can discriminate between sleep perceived as poor, moderate, and good. We hypothesized that certain sleep variables have greater weight in representing sleep quality and quantity, do not vary between players, and are highly associated with players’ perception of sleep.

## 2. Materials and Methods

### 2.1. Subjects

Twenty professional female players participated in this study (age: 23.3 ± 3.5 yrs; height: 169.7 ± 5.5 cm; body mass: 64.6 ± 5.3 kg) who play in the Spanish Women’s First Division (Liga Profesional Femenina). The inclusion criteria in the study were as follows: only players with more than 30 records were included in the analysis. The authorization of an institutional ethics committee was required for the collection of OURA^®^ (Oulu, Finland) data (C2TI0327). All players who participated in the study signed a document agreeing to the use of the OURA^®^ ring device. This study conformed to the Declaration of Helsinki, and players provided informed consent before participating; their identities were anonymized.

### 2.2. Objective Variables: Sleep Quantity and Quality

In line with previous studies [21,22] and incorporating some new ones, a total of 10 objective sleep variables were analyzed: EFFICENCY, REST, ONSET, DURA, TOTAL, AWAKE, REM, DEEP, LIGHT, and MIDP. Each variable represents a trait of sleep quality or quantity [7] and is defined as follows: EFFICENCY (%) is the percentage of the sleep period spent asleep; REST (%) is the percentage of sleep time during which the user was moving, where a higher value indicates increased movement during sleep; ONSET (hr:min) is the detected latency from the start of bedtime to the beginning of the first five minutes of persistent sleep; DURA (hr:min) is the total duration of the sleep period (DURA = bedtime end − bedtime start); TOTAL (hr:min) is the total amount of sleep registered during the sleep period (TOTAL = REM + LIGHT + DEEP); AWAKE (hr:min) is the total amount of awake time registered during the sleep period; REM (hr:min) is the total amount of REM sleep registered during the sleep period; DEEP (hr:min) is the total amount of deep sleep registered during the sleep period; LIGHT (hr:min) is the total amount of light sleep registered during the sleep period; MIDP (hr:min) is the time from the start of sleep to the midpoint of sleep based on the connected mobile device clock, ignoring awake periods.

### 2.3. Subjective Sleep Quality Variable

Subjective sleep quality (SSQ) was also recorded using the modified Hooper scale [23], where each player was asked each morning before the training session (among other questions) how she had slept that night (0–10 scale). The responses were then recategorized into three groups based on the player’s perception of the previous night’s sleep quality: poor (SSQ ≤ 5), moderate (SSQ 6 or 7), and good sleep (SSQ ≥ 8).

### 2.4. Design and Procedures

A longitudinal, observational design was used with sleep data collected during the pre-season and in-season periods of two consecutive seasons (2021–2022 and 2022–2023). Only players (n = 20) with more than 30 records were included in the analysis. A total of 2031 individual data records were analysed, with an average of 101.5 ± 72.9 records per player (minimum: 30, maximum: 248). The data included all days of the week with training sessions and matches, regardless of the player’s participation in the session. It also accounted for nights spent away from home during away games. Data from the OFF day of the week were not considered.

Sleep data was collected using a multisensor sleep tracker (ŌURA ring, Gen2, Oura, Oulu, Finland). The players wore ŌURA rings throughout the day, except during training sessions and matches, as regulations do not allow them to wear rings [24]. The rings are waterproof, made of ceramic, available in different sizes (U.S. standard ring sizes 6–13), and weigh about 15 g, with a battery life of approximately three days. The ring automatically connects via Bluetooth and transfers data to a mobile platform through the mobile app. Previous studies have demonstrated a high validity and reliability of the OURA ring by comparing the device with PSG, the gold standard, in the global variables such as TOTAL, DURA, and ONSET [25,26]. However, there is controversy regarding the variables LIGHT, DEEP, and REM. Svensson [25] report variability and reliability in the DEEP and LIGHT variables, while de Zambotti [26] suggest that OURA overestimates time spent in the REM phase and underestimates time spent in the DEEP phase. On the other hand, a previous study reported an accuracy of >90% in detecting wake/sleep phases using the same device employed in the current study [27], along with a low coefficient of variation of 10% [7].

### 2.5. Data Analysis

Prior to PCA, the dataset was examined for missing/erroneous data, which may have occurred due to OURA devices not being worn. In addition to providing absolute values for the variables, these devices also convert the values into a scale or score, which is more intuitive for users. Due to the high correlation between the absolute value and their corresponding scores, and to avoid issues with PCA [28], it was decided to include only the variables with their absolute values, thereby discarding variables with high correlations [29]. To avoid confounding variables (such as sleeping timing or environmental factors) that could interfere with the study, the players were asked to follow healthy habits. Additionally, the Bartlett test of sphericity and the Kaiser-Meyer-Olkin (KMO) measure of sampling adequacy were performed to assess the suitability of the data for PCA. The Bartlett test of sphericity was significant (x^2^ = 165,273, df = 45, *p* < 0.001) with a KMO above 0.5, suggesting that the data were suitable for PCA [30]. Eigenvalues > 2 were considered for PC extraction [31]. To account for the high correlation of the components and to ensure that each principal component provided different information, a Varimax Orthogonal Rotation method was used. For interpretation, the threshold on the loading of each PC was set at 0.6, with the highest factor loading being considered when cross-loading between components was observed. The qualitative descriptors for the magnitude of the correlations were as follows: <0.1 trivial; 0.1 to 0.3 small; 0.3 to 0.5 moderate; 0.5 to 0.7 large; 0.7 to 0.9 very large; 0.9 to 1.0 almost perfect [32].

A cluster analysis was performed to group similar sleep patterns. A K-means cluster analysis was conducted using the Euclidean distance metric. Prior to this, the data was standardized. The optimal number of clusters (k) was determined using the elbow method [33], indicating that k = 3 is the optimal inflection point, suggesting that three clusters provide an adequate structure for the data, capturing the main variability without overfitting the model.

Finally, to compare the means among the three identified clusters, we conducted a one-way analysis of variance (ANOVA) followed by a Bonferroni post hoc test to evaluate specific differences between the groups [34]. To evaluate the association between categorical variables (SSQ versus Cluster), a contingency table (3 × 2) was created with the observed frequencies. Then, the Chi-Square (x^2^) statistic was then calculated under the hypothesis of independence. Finally, the significance of the statistic was determined using a chi-square distribution table [35]. JASP version 0.18.3 (https://jasp-stats.org/, accessed on 16 May 2024) was used for statistical analysis.

## 3. Results

Table 1 shows the descriptive values (mean, standard deviation, minimum, and maximum) of the objective sleep variables for all the participants across all recorded sessions.

Figure 1 shows the correlation between all the objective sleep variables. The strongest correlations were observed between DURA with MIDP (r = 0.96; *p* < 0.001) and TOTAL (r = 0.91; *p* < 0.001), as well as between AWAKE and EFFICIENCY (r = −0.92; *p* < 0.001). Most variables had a moderate to very large association. Additionally, there were also some trivial or small correlations between sleep variables: REST and DEEP exhibited a small negative correlation with most of the other variables.

Table 2 details the PCA results for the sleep variables, with two dimensions explaining a cumulative variation of 72%. For the first PC (quantity PC) (45.1% of variance), the eigenvectors suggest a contribution from five of the total variables, with higher weights for TOTAL, DURA, and MIDP (0.93 to 0.98). For the second PC (quality PC) (26.5% of the variance), the eigenvectors indicate higher negative weights for EFFICIENCY (−0.94) and DEEP (−0.47), as well as positive weights for AWAKE and ONSET.

Variables highly related to PCs (considered significant if above 0.6) for each player are shown in Table 3. In general, similar results were found among the players, though some differences were observed in the variables ONSET, with a coefficient of variation (CV) of 24%, and REM (CV = 15%).

Table 4 shows the mean values (standard deviation) of the objective sleep variables for each cluster, as well as the significant differences. Cluster 3, which represents higher sleep quality, showed significant differences in most variables compared to the other two clusters. Cluster 1, despite having the longest total mean sleep duration (9:16), had the lowest mean sleep efficiency (80.7%) and the longest mean awake time (1:48). Cluster 2, with the shortest mean total sleep duration (6:53), showed intermediate efficiency (87.2%) and the shortest mean sleep onset time (00:11).

Table 5 shows the distribution of training sessions in relation to the cluster to the players’ sleep type (cluster) and the perception of sleep quality (SSQ) collected through the questionnaire. The chi-square test did not show a significant association between the two variables (x^2^ = 8.24, gl = 4, and *p* = 0.06).

The distribution shows that Cluster 2 (short time in bed but high efficiency) is the most common among players, accounting for 48% of the total observations (Table 6). However, there is considerable variability among individual players, with some showing a predominance in Cluster 1 (long time in bed but a lot of time AWAKE) or Cluster 3 (long time asleep with high EFFICIENCY).

## 4. Discussion

The primary aim of the present study was to uncover multivariate relationships within quantity and quality sleep variables to reduce the reporting of redundant data to coaches using PCA. Correlation analysis revealed strong relationships between multiple sleep variables, with subsequent PCA identifying two PCs explaining a combined 76% of the variance in sleep data. While similar within-player results were observed, some variation in variable loadings on each PC existed, suggesting that some individual player characteristics may have been hidden by analyzing the full dataset. The second objective was to examine whether the objective variables of quantity and quality of sleep can be associated with poor, moderate, or good sleep perceived by the players; however, no significant association was found between them (*p* = 0.08).

Correlation analysis revealed most variables had a moderate to very large association, although some correlations between sleep variables were trivial or small (Figure 1). In this sense, the highest correlations were observed between DURA and TOTAL, DURA and MIDP, and AWAKE and EFFICIENCY (with a negative correlation in the latter case). This highlights redundancy in the dataset, as many variables tend to change at similar rates across the different days studied, reinforcing the need for data reduction techniques to explore these relationships further.

PCA analysis has been applied to the study of external load variables in various sports such as basketball [36], rugby [13], and football [12,13,36,37] to examine different variables and facilitate their selection process [15]. To the authors’ knowledge, no previous studies have applied variable reduction techniques in the study of sleep data. This approach highlights the importance of evaluating sleep through different variables, selecting those that provide complementary information to aid decision-making [38]. It is crucial to consider both the quantity and quality of sleep and, ultimately, the quality of the quantity.

Total sleep time is sensitive to various external factors, such as the room type (single vs. double) [39], international travel [40], or light exposure [40]. The first PC (quantity PC) captures the most information, accounting for approximately 45% of the variance, with each subsequent PC a progressively smaller amount of variance. Within this quantity PC, several sleep variables closely related to sleep quantity contribute, with the TOTAL sleep variable having the greatest weight in quantity PC.

Efficiency (%) has been measured in different studies, finding it to be a variable with a low coefficient of variation (7%), being one of the recommended parameters for sleep monitoring [7]. Previous studies indicate that it is a variable affected by the type of room (single vs. double) in young soccer players, with significantly higher values in single rooms [39]. In the second of the components (quality PC), which captured ~26% of the additional sleep information, the variables appear to be related to sleep quality. Specifically, the EFFICIENCY variable carries the most weight in this PC.

When PCA was performed on individual players, similar results were found; however, some individual differences were observed. For example, in quantity PC, the variable REM (when the brain increases metabolic heat production and temperature) exhibited a variability (CV) of 15% between players. In quality PC, it was the ONSET variable (the time it takes the player from the onset of sleep to the start of the first five minutes of persistent sleep) that exhibited the highest inter-player variability, with a CV = 24%. While individual variation highlights a specific response to individual player demands (e.g., training and match loads, travel, personal life) and suggests the potential need to utilize particular sleep variables among players, selecting different variables for different players could be counterproductive. This approach would complicate coaches’ efforts to assess sleep quality and quantity in a comprehensive and sustainable manner.

Finally, cluster analysis enabled us to identify three groups of sleep types, revealing significant differences between them. Cluster 3 was characterized by higher sleep duration and sleep quality, while cluster 2, despite having a shorter total duration, exhibited a high sleep efficiency. In contrast, cluster 1 had the longest total duration but the lowest sleep efficiency. Several studies suggest that athletes tend to overestimate the quality and quantity of sleep, noting non-significant correlations between objective and subjective variables [18,19,20]. In this study, when the values of the perception of sleep quality collected through the questionnaire were compared to the objective variables grouped in the estimated clusters, a tendency but not an association (*p* = 0.06) was found between them, i.e., the objective variables did not correspond to the subjective variables perceived by the players. In this regard, it could be beneficial for players to receive training or educational talks on the importance of sleep. This might encourage them to take a greater interest in actively monitoring and analyzing their sleep. Additionally, it would be valuable to combine objective and subjective measures of sleep quality to cross-reference the data, thereby enhancing the validity of their perceptions and providing a clearer understanding of each individual’s sleep patterns.

This is the first study to address the combination of objective and subjective perspectives in monitoring sleep quality in such a large volume of records. However, some of the main limitations of the study include not having considered the type of session (match day approach) or microcycle in the analysis, as well as not analyzing sleep on the players’ days off. Another limitation was the failure to account for the differences in the chronotype characteristics of the players (e.g., evening types, morning types, and indifferent types), which could impact their ability to assimilate training loads and result in varying degrees of circadian disruption to the wake-sleep cycle [41]. Several studies have demonstrated the validity and reliability of the OURA ring for variables such as DURA, TOTAL, and ONSET in comparison to the gold standard PSG. However, the results are less clear for more sensitive variables such as DEEP, LIGHT, and REM. This uncertainty could affect the conclusions of the study if a sleep measurement device other than the OURA ring is used. Nevertheless, it remains unclear whether significant differences exist in these sensitive variables.

It is suggested that quantity and quality sleep have multidimensional measures; therefore, relying on only a single sleep variable could potentially lead to misunderstandings of sleep quality and quantity in female professional soccer players. Findings suggest that a combination of two sleep variables is required during sleep in female professional soccer players. Practitioners could quantify the sleep considering TOTAL, DURA, or MIDP, plus EFFICENCY, AWAKE, or DEEP.

Cluster analysis allowed grouping the players’ sleep into three types: long and efficient, short and efficient, and long and inefficient; however, no association was found between the subjective questionnaire and the sleep clusters. Consequently, the study underscores the imperative need to train female players in the process of subjective assessment of their sleep or to optimize the subjective questionnaire used, allowing it to serve as a more cost-effective and sustainable tool while providing reliable information on sleep quality. In this sense, it could be interesting for players to be able to combine the objective and subjective recording of sleep quality to contrast the information in order to optimize the validity of their perceptions.

## Figures and Tables

**Figure 1 sensors-25-00148-f001:**
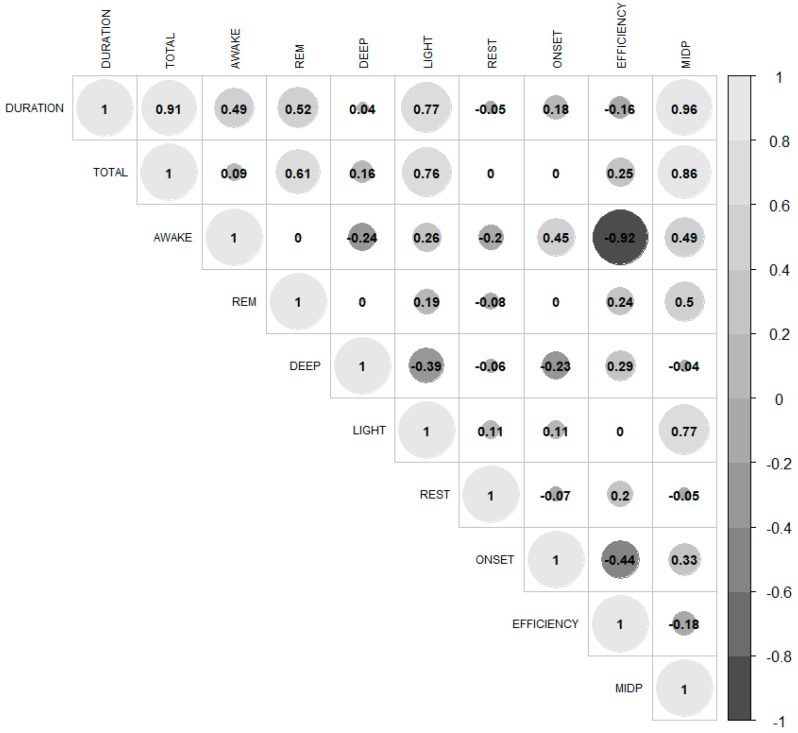
Correlation matrix for the sleep variables. Note: EFFICENCY (%) Is the percentage of the sleep period spent asleep; REST (%) is the percentage of sleep time during which the user was moving; ONSET (hr:min) is the detected latency from the start of bedtime to the beginning of the first five minutes of persistent sleep; DURA (hr:min) is the total duration of the sleep period; TOTAL (hr:min) is the total amount of sleep registered; AWAKE (hr:min) is the total amount of awake time registered; REM (hr:min) is the total amount of REM sleep registered; DEEP (hr:min) is the total amount of deep sleep registered; LIGHT (hr:min) is the total amount of light sleep registered; MIDP (hr:min) is the time from the start of sleep to the midpoint of sleep based on the connected mobile device clock, ignoring awake periods.

**Table 1 sensors-25-00148-t001:** Descriptive values (mean, standard deviation, minimum, and maximum) of the objective and subjective sleep variables.

Variables		Mean	Sd	Min	Max
DURA (hr:min)		08:30	01:19	03:35	15:00
TOTAL (hr:min)		07:20	01:09	03:20	11:23
AWAKE (hr:min)		01:10	00:32	00:11	05:13
REM (hr:min)		01:08	00:31	00:00	03:26
LIGHT (hr:min)		04:12	01:00	00:34	07:58
DEEP (hr:min)		02:00	00:34	00:16	03:53
MIDP (hr:min)		04:16	00:42	01:44	07:05
EFFICIENCY (%)		86.4	5.3	57.0	96.0
ONSET (hr:min)		00:14	00:12	00:02	02:07
REST (%)		8.50	8.37	3.0	49.0
SSQ (au)	Poor (n)		535		
	Moderate (n)		632		
	Good (n)		710		

Note: EFFICENCY (%) Is the percentage of the sleep period spent asleep; REST (%) is the percentage of sleep time during which the user was moving; ONSET (hr:min) is the detected latency from the start of bedtime to the beginning of the first five minutes of persistent sleep; DURA (hr:min) is the total duration of the sleep period; TOTAL (hr:min) is the total amount of sleep registered; AWAKE (hr:min) is the total amount of awake time registered; REM (hr:min) is the total amount of REM sleep registered; DEEP (hr:min) is the total amount of deep sleep registered; LIGHT (hr:min) is the total amount of light sleep registered; MIDP (hr:min) is the time from the start of sleep to the midpoint of sleep based on the connected mobile device clock, ignoring awake periods. SSQ—subjective sleep quality: poor (SSQ ≤ 5), moderate (SSQ 6 or 7), and good sleep (SSQ ≥ 8).

**Table 2 sensors-25-00148-t002:** Principal component (PC) analysis results for the sleep data showing eigenvalues, percentage of variance explained, cumulative variance explained, and component loadings for the three PCs.

	Quantity PC	Quality PC
Eigenvalue	4.1	2.4
% of total variance explained (VE)	45.1	26.4
Cumulative % of total VE	45.1	71.5
TOTAL	0.98	
DURA	0.95	
MIDP	0.93	
LIGHT	0.79	
REM	0.65	
EFFICIENCY		−0.94
AWAKE		0.91
ONSET		0.62
DEEP		−0.47

Note: EFFICENCY (%) Is the percentage of the sleep period spent asleep; REST (%) is the percentage of sleep time during which the user was moving; ONSET (hr:min) is the detected latency from the start of bedtime to the beginning of the first five minutes of persistent sleep; DURA (hr:min) is the total duration of the sleep period; TOTAL (hr:min) is the total amount of sleep registered; AWAKE (hr:min) is the total amount of awake time registered; REM (hr:min) is the total amount of REM sleep registered; DEEP (hr:min) is the total amount of deep sleep registered; LIGHT (hr:min) is the total amount of light sleep registered; MIDP (hr:min) is the time from the start of sleep to the midpoint of sleep based on the connected mobile device clock, ignoring awake periods.

**Table 3 sensors-25-00148-t003:** Correlations between sleep variables and the two first PCs (quantity PC and quality PC) for individual players.

Player	Quantity PC					Quality PC			
	TOTAL	DURA	MIDP	LIGHT	REM	EFFICIENCY	AWAKE	ONSET	DEEP
1	0.98	0.99	0.98	0.84	0.57	−0.86	0.94	0.68	−0.83
2	0.99	0.99	0.98	0.9	0.7	−0.88	0.97	0.63	−0.54
3	0.97	0.99	0.97	0.81	0.73	−0.9	0.95	0.82	−0.85
4	0.97	0.96	0.98	0.56	0.71	−0.92	0.94	0.46	−0.93
5	0.97	0.99	0.98	0.83	0.64	−0.9	0.96	0.4	−0.89
6	0.99	0.97	0.98	0.89	0.88	−0.77	0.95	0.38	−0.91
7	0.97	0.99	0.97	0.93	0.81	−0.85	0.97	0.57	−0.82
8	0.9	0.97	0.95	0.85	0.45	−0.92	0.96	0.48	−0.89
9	0.95	0.97	0.95	0.86	0.72	−0.89	0.95	0.61	−0.89
10	0.97	0.96	0.97	0.82	0.77	−0.87	0.93	0.63	−0.91
11	0.95	0.96	0.97	0.72	0.71	−0.95	0.96	0.58	−0.94
12	0.98	0.99	0.98	0.76	0.7	−0.83	0.92	0.72	−0.92
13	0.97	0.99	0.96	0.85	0.71	−0.89	0.93	0.69	−0.9
14	0.94	0.99	0.94	0.82	0.65	−0.87	0.94	0.77	−0.9
15	0.98	0.98	0.98	0.76	0.63	−0.87	0.93	0.41	−0.93
16	0.99	0.99	0.97	0.85	0.84	−0.81	0.92	0.82	−0.91
17	0.94	0.98	0.95	0.7	0.7	−0.9	0.96	0.87	−0.95
18	0.98	0.98	0.98	0.79	0.89	−0.85	0.96	0.59	−0.96
19	0.99	0.98	0.98	0.76	0.8	−0.74	0.82	0.5	−0.81
20	0.86	0.96	0.81	0.82	0.69	−0.95	0.96	0.78	−0.88
CV	3%	1%	4%	10%	15%	7%	7%	24%	10%

Note: EFFICENCY (%) Is the percentage of the sleep period spent asleep; REST (%) is the percentage of sleep time during which the user was moving; ONSET (hr:min) is the detected latency from the start of bedtime to the beginning of the first five minutes of persistent sleep; DURA (hr:min) is the total duration of the sleep period; TOTAL (hr:min) is the total amount of sleep registered; AWAKE (hr:min) is the total amount of awake time registered; REM (hr:min) is the total amount of REM sleep registered; DEEP (hr:min) is the total amount of deep sleep registered; LIGHT (hr:min) is the total amount of light sleep registered; MIDP (hr:min) is the time from the start of sleep to the midpoint of sleep based on the connected mobile device clock, ignoring awake periods. CV—coefficient of variation. The darker the cell, the higher the values within each variable.

**Table 4 sensors-25-00148-t004:** Descriptive values (means and standard deviation) of the objective sleep variables for each cluster.

Variable	Cluster		
	1	2	3
DURA	09:16 (00:58) ^2,3^	06:53 (01.00)	08:59 (00:45) ^2^
TOTAL	07:28 (00:52) ^2^	06:00 (00:55)	08:00 (00:42) ^2,1^
AWAKE	01:48 (00:31) ^2,3^	00:53 (00:20)	00.59 (00:16)
REM	01:06 (00:27) ^2^	00:45 (00:23)	01:22 (00:29) ^1,2^
LIGHT	04:39 (00:48) ^2^	03:10 (00:45)	04:31 (00:47) ^2^
DEEP	01:43 (00:29)	02:05 (00:37) ^1^	02:07 (00:32) ^1^
MIDP	04:42 (00:33) ^2^	03:24 (00:30)	04:29 (00:24) ^2^
EFFICIENCY	80.7 (4.9)	87.2 (4.5) ^1^	89.2 (2.8) ^1^
ONSET	00:24 (00:17) ^2,3^	00:11 (00:08)	00:11 (00:07)
REST	6.6 (3.4)	8.6 (7.9) ^1^	9.6 (10.3) ^1^

Note: EFFICENCY (%) Is the percentage of the sleep period spent asleep; REST (%) is the percentage of sleep time during which the user was moving; ONSET (hr:min) is the detected latency from the start of bedtime to the beginning of the first five minutes of persistent sleep; DURA (hr:min) is the total duration of the sleep period; TOTAL (hr:min) is the total amount of sleep registered; AWAKE (hr:min) is the total amount of awake time registered; REM (hr:min) is the total amount of REM sleep registered; DEEP (hr:min) is the total amount of deep sleep registered; LIGHT (hr:min) is the total amount of light sleep registered; MIDP (hr:min) is the time from the start of sleep to the midpoint of sleep based on the connected mobile device clock, ignoring awake periods. ^1^ is >cluster 1, ^2^ is >cluster 2, and ^3^ is >cluster 3, at *p* < 0.05.

**Table 5 sensors-25-00148-t005:** Distribution of the training sessions in relation to clusters and subjective perception of sleep quality (SSQ).

SSQ	Cluster			Total
	1	2	3	
Good	172	332	206	710
Moderate	158	330	144	632
Poor	142	245	148	535
Total	472	907	498	1877

Note: sleep quality: poor (SSQ ≤ 5), moderate (SSQ 6 or 7), and good sleep (SSQ ≥ 8).

**Table 6 sensors-25-00148-t006:** Distribution of players’ records in the three clusters.

Player	Cluster			
	1	2	3	Total
1	46 (21.6%)	135 (63.4%)	32 (15.0%)	213
2	2 (2.9%)	46 (66.7%)	21 (30.4%)	69
3	5 (6.2%)	55 (67.9%)	21 (25.9%)	81
4	1 (3.3%)	18 (60.0%)	11 (36.7%)	30
5	4 (10.3%)	24 (61.5%)	11 (28.2%)	39
6	11 (34.4%)	8 (25.0%)	13 (40.6%)	32
7	31 (34.8%)	43 (48.3%)	15 (16.8%)	89
8	27 (60.0%)	7 (15.6%)	11 (24.4%)	45
9	123 (59.1%)	42 (20.2%)	43 (20.7%)	208
10	22 (25.3%)	52 (59.8%)	13 (14.9%)	87
11	16 (25.0%)	34 (53.1%)	14 (21.9%)	64
12	14 (6.1%)	104 (45.4%)	111 (48.5%)	229
13	78 (31.4%)	139 (56.0%)	31 (12.5%)	248
14	67 (50.4%)	34 (25.6%)	32 (24.1%)	133
15	17 (9.1%)	77 (41.4%)	92 (49.5%)	186
16	4 (5.5%)	50 (68.5%)	19 (26.0%)	73
17	7 (13.5%)	19 (36.5%)	26 (50.0%)	52
18	12 (21.4%)	35 (62.5%)	9 (16.1%)	56
19	0 (0.0%)	46 (80.7%)	11 (19.3%)	57
20	25 (62.5%)	9 (22.5%)	6 (15.0%)	40
Total	512 (25.2%)	977 (48.1%)	542 (26.7%)	2031

## Data Availability

Data are contained within the article.

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
