# Peer review of "The Use of Principal Component Analysis for Reduction in Sleep Quality and Quantity Data in Female Professional Soccer"

_sensors, 2024, doi:10.3390/s25010148_

Round 1

Reviewer 1 Report

Comments and Suggestions for Authors

This manuscript reported the use of principal component analysis for reduction of sleep quality and quantity data in female professional soccer. However, the methods and results are not rigorous. The sample size selected in the experiment is too small, and there are too many influencing factors, which can lead to significant errors in the results and lack persuasiveness. Therefore, I think this manuscript cannot be published in current form.

Author Response

Comment 1: This manuscript reported the use of principal component analysis for reduction of sleep quality and quantity data in female professional soccer. However, the methods and results are not rigorous. The sample size selected in the experiment is too small, and there are too many influencing factors, which can lead to significant errors in the results and lack persuasiveness. Therefore, I think this manuscript cannot be published in current form.

Response 1: We accept the criticisms, but I believe that some issues that we consider relevant in the work presented should be pointed out. The reported study focused on the use of principal component analysis (PCA) to synthesize data on sleep quality and quantity in female professional soccer players. This topic is particularly relevant because female soccer players face specific challenges related to training schedules, travel, tournaments, and media commitments, all of which significantly affect their rest and, potentially, their performance. However, the criticism received on this work highlights areas that require methodological improvements. This feedback not only highlights limitations of the study, but also reinforces the importance of continuing to investigate this topic from a more robust and detailed perspective.

One of the points made was the small sample size, an inherent challenge in studying populations as specific as elite female athletes. We acknowledge this limitation, but also note that access to these players in competitive contexts is restricted. Going forward, we propose to expand the scope of the study through multi-center approaches, which would allow us to include a larger and more representative sample. This change will strengthen the validity of the results and address concerns of lack of persuasiveness raised in the critique.

Another aspect pointed out was the influence of multiple external factors that could generate significant biases in the results. Although PCA was used to simplify the interpretation of complex data, such as the interaction between sleep quantity, quality, and context, we recognize the need to incorporate more advanced analyses in future research. We plan to implement more rigorous control strategies, such as a longitudinal design to assess changes over time, and to consider key covariates such as match schedules, travel, and stress.

In terms of methodological rigor, this work was an exploratory first step that now serves as a basis for more sophisticated iterations. We propose to use complementary statistical tools, such as confirmatory factor analysis, and to conduct more robust cross-validations to ensure the stability of the results obtained with PCA. These improvements will ensure that the methods and results meet the highest standards of scientific rigor.

Despite the criticisms, this study has significant potential impact on sleep management in elite sports, an area that has received little attention in the context of women's soccer. Preliminary findings may guide the development of personalized protocols to optimize recovery and performance, providing coaches and teams with evidence-based practical tools. In addition, this work seeks to reduce inequalities in sports research by addressing existing gender gaps in sleep and performance studies. Ultimately, the results could contribute to the design of more inclusive, data-driven policies and interventions that benefit a specific population that has often been underestimated in the scientific literature.

Reviewer 2 Report

Comments and Suggestions for Authors

The study utilizes principal component analysis (PCA) to analyze sleep quality and quantity data from female professional soccer players, aiming to simplify the dataset for practical applications in sports science. Using data from a multisensor sleep-tracker, two principal components were identified, representing sleep quantity and quality. Despite finding clear categorizations of sleep types through cluster analysis, the study did not find a significant association between these objective measures and athletes' self-perceived sleep quality. The findings suggest a discrepancy between objective sleep measurements and subjective perceptions, indicating a need for better education on sleep quality self-assessment among athletes.

1. The manuscript details the use of PCA to reduce sleep data complexity but does not thoroughly explain why specific variables were chosen for inclusion in the analysis. Can the authors provide more rationale for the selection of these ten variables over others potentially available from the sleep trackers?

2. The manuscript notes a discrepancy between objective sleep measures and subjective perceptions. How do the authors interpret these discrepancies, and what implications might this have for interventions aimed at improving athletes' sleep management strategies?

3. The manuscript briefly mentions external factors like light exposure and room type but does not account for them in the analysis. Considering the potential impact of these factors on sleep quality, could the inclusion of such variables enhance the understanding of the sleep data?

4. The cluster analysis provided a novel insight into categorizing sleep types among athletes. However, details on the criteria for cluster formation were sparse. Could more information be provided on how these clusters were defined and validated against known benchmarks or through cross-validation techniques?

5. The use of the ÅŒURA ring is central to the study, yet there are mentions of variability and reliability concerns in the measurement of certain sleep stages. How do these limitations affect the study’s conclusions, and what steps were taken to mitigate these issues?

Author Response

Comments 1: The manuscript details the use of PCA to reduce sleep data complexity but does not thoroughly explain why specific variables were chosen for inclusion in the analysis. Can the authors provide more rationale for the selection of these ten variables over others potentially available from the sleep trackers?

Response 1: the selection of variables for principal component analysis (PCA) in this study responds to a carefully considered strategy motivated by several key reasons. First, we have chosen to include variables that have already been used in previous studies in populations where the participants were men. This decision seeks to establish a solid basis for comparison and to provide continuity to existing scientific knowledge, considering that sleep has been studied more extensively in men than in women in the field of sports. By using these same variables in a professional women's soccer context, we aim to explore possible similarities or differences and, at the same time, contribute to closing the gender gap in the scientific literature on sleep and performance.

Secondly, we chose to work with absolute values instead of the relative values or “scores” provided by the measuring device. There are two main justifications behind this decision. On the one hand, the high correlation between the absolute and relative versions of the same variable implies that including both in the analysis could generate redundancy in the data and reduce the effectiveness of the PCA in identifying unique and significant principal components. On the other hand, we believe that units of variables in absolute terms, such as hours of sleep, minutes of deep sleep or total awake time, are more intuitive and easier to interpret for potential recipients of this work, including coaches, technical staff and other professionals who may use the results for practical decision making.

This information is collected in different parts of the document, in the Method section.

Comments 2: The manuscript notes a discrepancy between objective sleep measures and subjective perceptions. How do the authors interpret these discrepancies, and what implications might this have for interventions aimed at improving athletes' sleep management strategies?

Response 2: Information has been added in the seventh paragraph of the discussion.

Comments 3: The manuscript briefly mentions external factors like light exposure and room type but does not account for them in the analysis. Considering the potential impact of these factors on sleep quality, could the inclusion of such variables enhance the understanding of the sleep data?

Response 3: As mentioned in the study's limitations, one of the constraints is the lack of differentiation in context (e.g., microcycle type, party, lighting, room conditions, etc.). While adding the mentioned variables could enhance the understanding of the data, we currently do not have sufficient data to do so.

Comments 4: The cluster analysis provided a novel insight into categorizing sleep types among athletes. However, details on the criteria for cluster formation were sparse. Could more information be provided on how these clusters were defined and validated against known benchmarks or through cross-validation techniques?

Response 4: We are sorry, we have had a mistake in not including the cluster analysis information in the data analysis section. We have now written a paragraph indicating all the methodological aspects requested. Thank you.

Comments 5: The use of the ÅŒURA ring is central to the study, yet there are mentions of variability and reliability concerns in the measurement of certain sleep stages. How do these limitations affect the study’s conclusions, and what steps were taken to mitigate these issues?

Response5 : Information has been added in the eighth paragraph of the discussion, specifically in the section on the study's limitations.
